# Blue organic light-emitting diode with a turn-on voltage of 1.47 V

Seiichiro Izawa ®[1,2,3] ✉, Masahiro Morimoto ®[4] ✉, Keisuke Fujimoto ®[5] ✉, Koki Banno[5], Yutaka Majima ®[1], Masaki Takahashi ®[5], Shigeki Naka ®[4] & Masahiro Hiramoto[6]

Among the three primary colors, blue emission in organic light-emitting diodes (OLEDs) are highly important but very difficult to develop. OLEDs have already been commercialized; however, blue OLEDs have the problem of requiring a high applied voltage due to the high-energy of blue emission. Herein, an ultralow voltage turn-on at 1.47 V for blue emission with a peak wavelength at 462 nm (2.68 eV) is demonstrated in an OLED device with a typical blue-fluorescent emitter that is widely utilized in a commercial display. This OLED reaches 100 cd/m$^2$, which is equivalent to the luminance of a typical commercial display, at 1.97 V. Blue emission from the OLED is achieved by the selective excitation of the low-energy triplet states at a low applied voltage by using the charge transfer (CT) state as a precursor and triplet-triplet annihilation, which forms one emissive singlet from two triplet excitons.

Blue is the most important constituent color in light-emitting devices because it has the highest energy among the three primary colors in displays, and white emission in lighting applications is made by a blue light source[1]. Organic light-emitting diodes (OLEDs) have already been commercialized in smartphones and large-screen displays, exploiting their ability to project high-color images with large contrast[2]. However, blue OLEDs still have drawbacks because of the need for a large applied voltage because the energy of blue emission can be as high as approximately 3 eV. Typical blue OLEDs need approximately 4 V for a luminance of 100 cd/m$^2$, which is a general display condition[3]. The industrial target is to operate blue OLEDs within 3.7 V, which is the rated voltage of the lithium-ion batteries that are loaded in most mobile devices. For this reason, blue OLEDs operating at low voltages are highly desirable for achieving commercial requirements.

Conventional fluorescent emitters are still used in commercial blue OLEDs due to their reliability and long operation lifetime[4], although their external quantum efficiencies (EQEs) in devices are lower than those of phosphorescent and thermally activated delayed

fluorescence (TADF) materials; devices with phosphorescent and TADF materials are emerging technologies in academia[5,6]. The energy of the first triplet excited state (T$_1$) of anthracene derivatives, which are some of the most typical fluorescent emitters, is stable at 1.7 eV[7]. In contrast, blue phosphorescent and TADF materials have a T$_1$ that can be as high as 3 eV[4]. High energy levels are inevitable when considering their operating mechanism: the T$_1$ energies must be equal to or close to the blue light energy in the phosphorescent or TADF material. Here, the spin forbidden T$_1$ has a long lifetime, and 3 eV is equivalent to the dissociation energy of a carbon-nitrogen bond[8]. Therefore, the high-energy T$_1$ will promote the degradation of the material. This intrinsic problem prevents the commercialization of phosphorescent and TADF materials for blue OLEDs, although several phosphorescent and TADF materials with relatively high stability have been reported recently[8–10].

The operation mechanism of a conventional fluorescent OLED is illustrated in Fig. 1a. The formation ratios of the first singlet excited state (S$_1$) and T$_1$ are 25% and 75%, respectively, due to the spin statistic rule[2]. An applied voltage ($V_{appl}$) multiplied by the elementary charge ($e$)

[1]Laboratory for Materials and Structures, Institute of Innovative Research, Tokyo Institute of Technology, 4259 Nagatsuta-cho, Midori-ku, Yokohama, Kanagawa 226-8503, Japan. [2]Joining and Welding Research Institute, Osaka University, 11-1, Mihogaoka, Ibaraki, Osaka 567-0047, Japan. [3]Precursory Research for Embryonic Science and Technology (PRESTO), Japan Science and Technology Agency (JST), 4-1-8 Honcho, Kawaguchi, Saitama 332-0012, Japan. [4]Academic Assembly Faculty of Engineering, University of Toyama, 3190 Gofuku, Toyama 930-8555, Japan. [5]Department of Applied Chemistry, Faculty of Engineering, Shizuoka University, 3-5-1 Johoku, Naka-ku, Hamamatsu, Shizuoka 432-8561, Japan. [6]Institute for Molecular Science, 5-1 Higashiyama, Myodaiji, Okazaki, Aichi 444-8787, Japan. ✉e-mail: izawa.s.ac@m.titech.ac.jp; morimoto@eng.u-toyama.ac.jp; fujimoto.keisuke@shizuoka.ac.jp

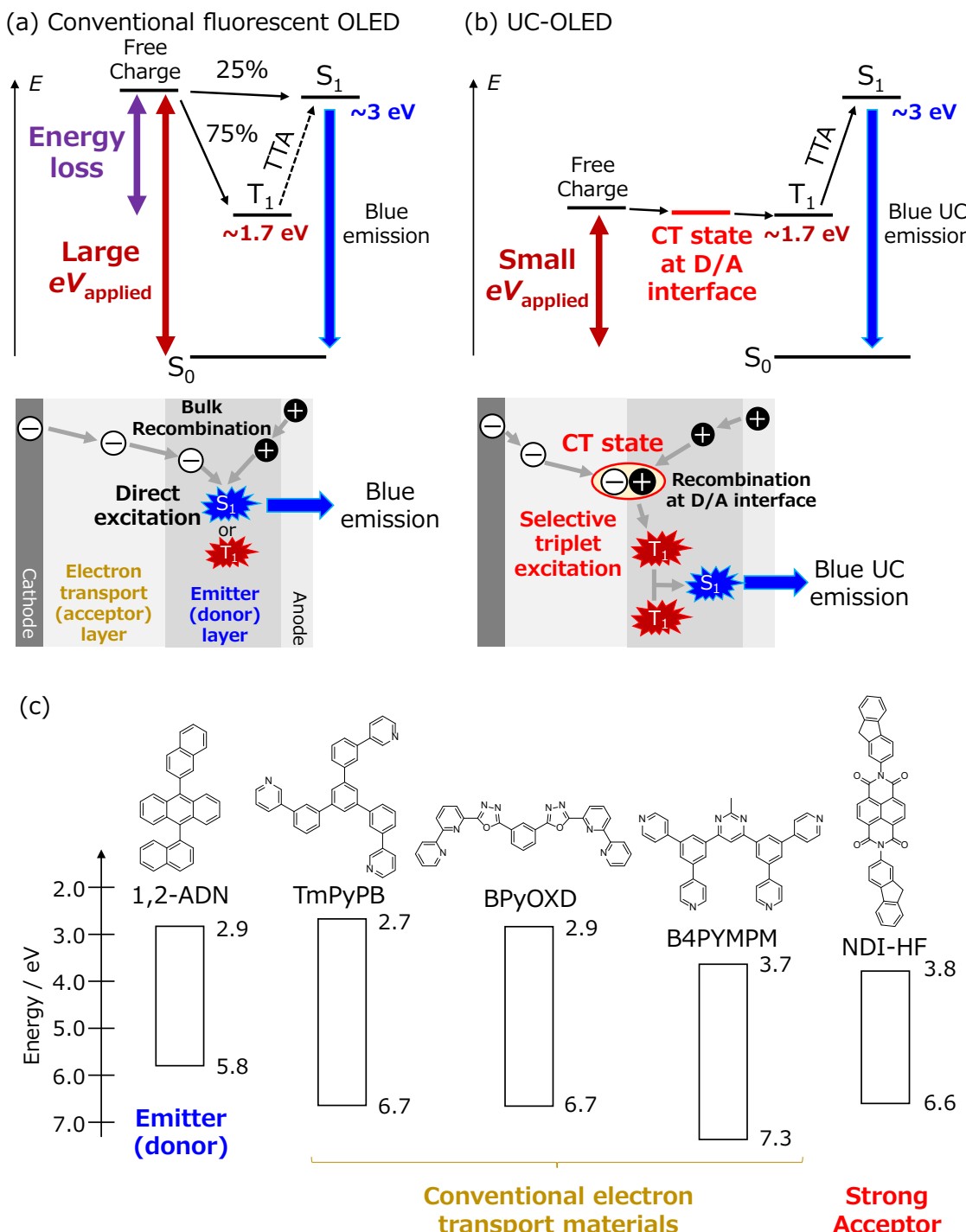

**Fig. 1 | Design principle of UC-OLED.** Schematic of the operating mechanism and the device structure of a conventional blue fluorescent OLED (**a**) and a blue UC-OLED (**b**). (**c**) Chemical structures and energy levels of the materials.

that is more than the bandgap energy of the emitter is needed to excite both the high-energy $S_1$ and low-energy $T_1$. This is because electrons and holes are independently injected from the charge transporting layers into the highest occupied molecular orbital (HOMO) and lowest unoccupied molecular orbital (LUMO) levels of the emitters, and the exciton binding energy and the extra energy related to the difference between $S_1$ and $T_1$ are lost to form $T_1$. A certain percentage of dark $T_1$ contributes to fluorescence through the triplet-triplet annihilation (TTA) process[4]. Therefore, we assumed that selective excitation of a low energy $T_1$ in a conventional emitter and subsequent TTA-based

fluorescence with high efficiency enabled a dramatic reduction in the applied voltage.

The operational mechanism of the device, which we call an upconversion (UC)-OLED, is illustrated in Fig. 1b. Initially, holes and electrons are injected into donor (emitter) and acceptor (electron transport) layers, respectively, and recombine at the donor/acceptor (D/A) interface to form a charge transfer (CT) state. Subsequently, the energy of the CT state is transferred to $T_1$ of the emitter. Therein, blue light is emitted through the formation of high-energy $S_1$ by TTA. Due to the much lower energy of the CT state than the bandgap energy of the

emitter, the UC process through TTA greatly reduces the applied voltage ($V_{appl}$) for exciting the emitter molecule compared to the conventional blue fluorescent OLED. The TTA-UC emission sensitized by the CT state has been mainly studied with rubrene, which shows yellow emission in previous studies[11–15]. In this study, we discovered an efficient blue TTA-UC emission phenomenon by determining an appropriate combination of blue emitter (donor) and acceptor (electron transport) from 21 organic molecules. According to this design concept, the turn-on voltage of the blue OLED with a typical fluorescent emitter that is widely utilized in a commercial display is greatly reduced to as low as 1.47 V, and the OLED reaches 100 cd/m², which is equivalent to the luminance of a typical display, at 1.97 V. Furthermore, the blue emission in the UC-OLED originated from a stable low energy $T_1$ and subsequent fast TTA-UC emission[16], which could potentially avoid degrading the constituent materials.

## Results

We utilized the anthracene derivative 1,2-ADN (9-(naphthalen-1-yl)-10-(naphthalen-2-yl)anthracene), which is one of the most widely used host materials in blue fluorescent OLEDs[17], as the emitter (donor) in our UC-OLEDs. Anthracene derivatives are also utilized as TTA emitters in the research field of photoexcited UC because they satisfy the energy requirements for efficient TTA: the energy of $T_1$ (1.7 eV) is slightly more than half the energy of $S_1$ (2.9 eV)[7,18,19]. As a partner to form the D/A interface with 1,2-ADN, we investigated two phenyl pyridine derivatives (TmPyPB and B4PYMPM)[20,21] and a bipyridyl-substituted oxadiazole derivative (BPyOXD)[22], which are typical electron transport materials in conventional OLED devices, and a naphthalene diimide derivative with a fluorene side chain (NDI-HF)[23]. NDI derivatives have strong electron acceptability; therefore, they are used as electron acceptors in the organic photovoltaic (OPV) field[24]. The energy levels of the materials in thin films have been evaluated by photoelectron spectroscopy and absorption spectra in previous studies, and the values are listed in Fig. 1c[20–23]. The HOMO levels of the electron transport materials are deeper than that of 1,2-ADN, whereas the LUMOs of electron transport (acceptor) materials lie in the order of TmPyPB, BPyOXD, B4PYMPM, and NDI-HF. We fabricated bilayer-type OLED devices with anthracene emitters and electron transport (acceptor) materials. The details of the device fabrication are described in the Methods section.

The device structure and properties of the OLED with 1,2-ADN and the four types of acceptor materials are exhibited in Fig. 2. Blue emission with a peak wavelength of 424 nm (2.92 eV) from 1,2-ADN is observed in all devices (Fig. 2b); however, the luminance-voltage ($L$-$V$) characteristics in Fig. 2c largely shift when different electron transport (acceptor) materials are used. The turn-on voltage, i.e., the voltage at which the electroluminescence (EL) emission reaches 1 cd/m²[14], is 3.5 V, 2.9 V, 2.8 V, and 1.7 V for TmPyPB, BPyOXD, B4PYMPM, and NDI-HF, respectively. Blue light can be emitted from approximately half the voltage of the photon energy in the 1,2-ADN/NDI-HF device[11,12,14]. This ultralow turn-on of blue emission is also observed when we use other anthracene derivatives and NDI-HF, as described in Fig. S1. The EQE of the OLED with NDI-HF is lower than that of other electron transport materials, as shown in Fig. 2e.

To clarify the origin of the different turn-on voltages in these devices, we investigated the decay dynamics of EL emission (Fig. 2f). There is a clear difference in the transient EL signals between these devices using the different electron transport (acceptor) materials. The OLED devices with typical electron transport materials (TmPyPB, BPyOXD, and B4PYMPM) mainly show prompt decay, with EL lifetimes of approximately 0.2 μs (which is the detection limit of the instrument). The prompt decay is due to the fast emission from the $S_1$ states that form directly after charge injection to the emitter layer, as illustrated in Fig. 1a[25]. The amplitude shows that approximately 80 - 90% of the EL emission is governed by fast emission in devices with the

three typical electron transport materials. In contrast, only a slow decay component with a lifetime on the order of μs is observed in the EL decay curves of 1,2-ADN/NDI-HF. The slow decay can be associated with emission originating from the TTA of the triplet excitons, which is a slow diffusion process[25]. The fitted exponential curve in Fig. S2 reveals that the lifetime of TTA emission in 1,2-ADN/NDI-HF is 0.85 μs, which is faster than the value in the UC-OLED with rubrene in a previous report (> 3 μs)[14], indicating faster triplet diffusion in 1,2-ADN than in rubrene. The results indicate that all of the emission in the 1,2-ADN/NDI-HF device is produced by TTA-UC and that the low energy $T_1$ of 1,2-ADN is selectively generated by charge injection (Fig. 1b). The energy levels of $S_1$ and $T_1$ of 1,2-ADN are 2.9 and 1.7 eV, respectively[7]. The energy difference of the initial excited state produced by the injected charge results in the difference in turn-on voltage between the OLED with NDI-HF and other electron transport materials.

Note that the energy difference of LUMO between B4PYMPM and NDI-HF is only 0.1 eV; however, the turn-on voltage and the operation mechanism of the devices are clearly different. To elucidate the origin of the difference, we investigated the highly sensitive incident photon-to-current conversion efficiency (IPCE) to measure the CT state absorption, which reflected the D/A interaction at the interface between the emitter (donor) and the electron transport (acceptor) materials in Fig. 2g[26,27]. The device with TmPyPB, BPyOXD, and B4PYMPM, which are typical electron transport materials in OLEDs, has little photocurrent response at wavelengths longer than the HOMO-LUMO transition of 1,2-ADN at approximately 450 nm. In contrast, the device with NDI-HF shows a clear photocurrent response until 700 nm, which is the signal of CT state formation at the D/A interface due to the strong interaction between 1,2-ADN and NDI-HF[28]. The results indicate that CT state formation is necessary for the direct excitation of $T_1$ at low voltage because the CT state acts as a precursor for energy transfer to $T_1$, as illustrated in Fig. 1b. Why NDI-HF is a special to form the CT state and show the TTA-UC emission is an intriguing question. We measured the XRD patterns of B4PYMPM, BPyOXD, TmPyPB, and NDI-HF in Fig. S3. There are no clear diffraction peaks in the XRD patterns of B4PYMPM, BPyOXD, and TmPyPB, indicating that these conventional electron transport materials are amorphous in the thin films due to weak intermolecular interactions. In contrast, NDI-HF shows diffraction peaks, and the intermolecular interaction of NDI-HF is too strong to dissolve in any organic solvent. The strong intermolecular interaction of NDI-HF leads to a large heteromolecular interaction with 1,2-ADN at the interface; therefore, only NDI-HF forms a CT state at the interface.

The next aspect is to determine if NDI-HF is the best acceptor material as the partner of 1,2-ADN for achieving efficient energy transfer from the CT state to $T_1$ of 1,2-ADN and consequently efficient TTA-UC emission. Therefore, we synthesized 14 NDI derivatives with a different substituent on the nitrogen position of NDI, as shown in Fig. 3a. Details of the synthesis and the material properties are summarized in the Supplementary Information. The substituents are broadly classified as aryl and alkyl groups. Table S1 summarizes the LUMO energy levels of the NDI derivatives. Most NDIs with aryl groups have lower LUMO levels than NDIs with alkyl groups, as reflected by the difference in the electron-donating properties from the substituents to the NDI core. The difference in the LUMO levels of NDI derivatives results in large differences in the EL emission spectra of the UC-OLEDs. Fig. 3b shows a typical example; the UC-OLED with NDI-HF containing an aryl group shows mostly TTA-UC emission at 450 nm, while the UC-OLED with NDI bearing a cyclohexyl group shows a clear CT emission at 665 nm (1.86 eV) with suppressed TTA-UC emission. The energy transfer scheme of the UC-OLED is illustrated in Fig. 3c. TTA-UC emission occurs through energy transfer from the triplet CT state ($CT_3$) to $T_1$ of 1,2-ADN[14]. However, a direct decay path from the CT state to the ground state via either radiative or nonradiative transition also exists. To determine the relationship between the efficiency of TTA-UC

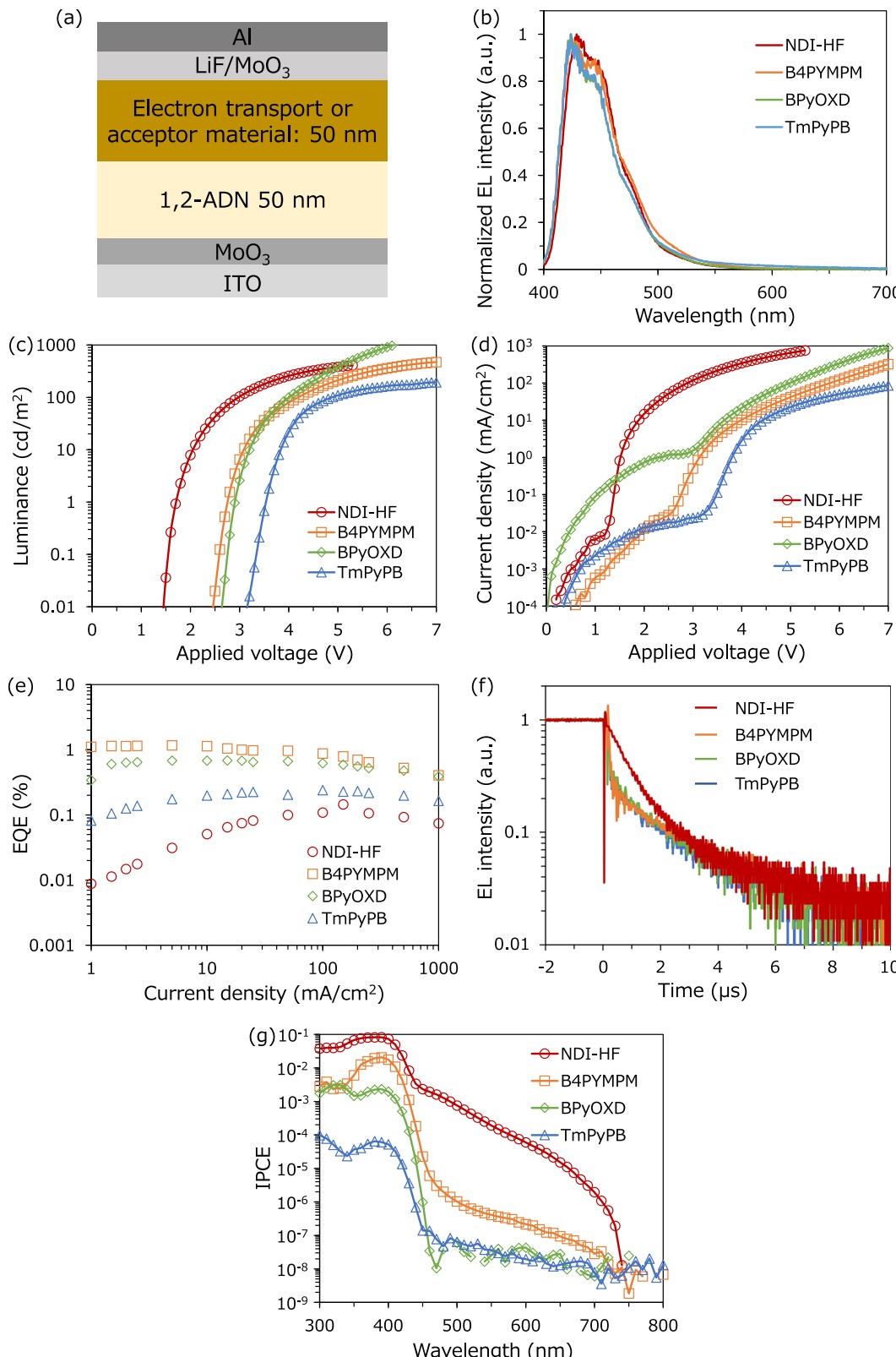

**Fig. 2 | Optical properties of UC-OLED. a** Schematic of the device structure. **b** EL emission spectra of the 1,2-ADN/NDI-HF (red circle), 1,2-ADN/B4PYMPM (orange square), 1,2-ADN/BPyOXD (green diamond) and 1,2-ADN/TmPyPB (blue triangle) devices under a constant current flow (100 mA/cm²). **c** L−V and **d** J-V curves and **e** EQE of the devices. **f** Decay dynamics of EL emission of the devices. Voltages of 3.5, 6.0, 8.0, and 8.0 V were applied to the 1,2-ADN/NDI-HF, 1,2-ADN/B4PYMPM, 1,2-ADN/BPyOXD and 1,2-ADN/TmPyPB devices, respectively. **g** Highly sensitive IPCE spectra of the devices.

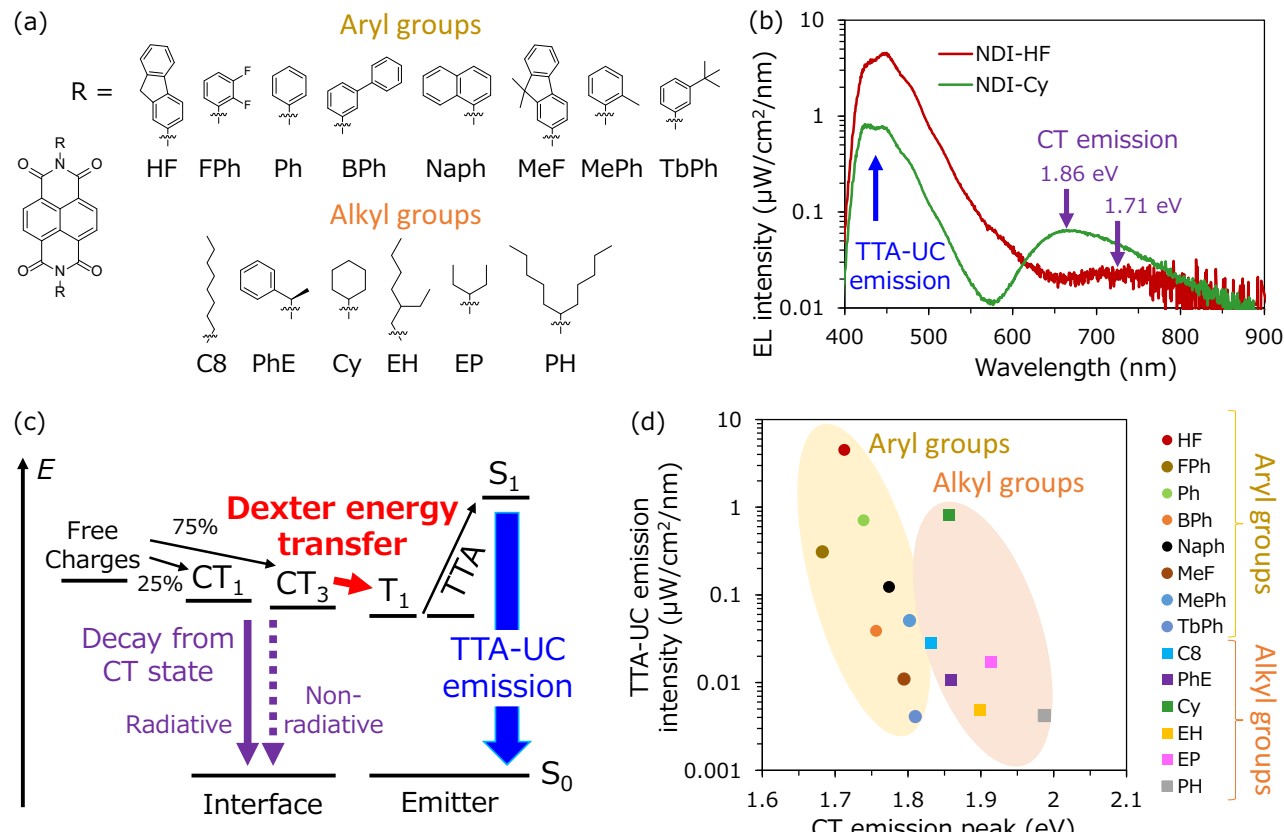

**Fig. 3 | Effect of molecular structure of acceptor. a** Chemical structures of NDI derivatives. **b** EL emission spectra of the 1,2-ADN/NDI-HF (red) and 1,2-ADN/NDI-Cy (green) devices under a constant current flow (100 mA/cm²). **c** Schematic of the energy transfer inside the UC-OLED. **d** Plots of the TTA-UC emission intensity versus energy of the CT emission peak for 1,2-ADN/NDI derivative devices.

emission and CT state energy, the TTA-UC emission intensity at constant current flow is plotted as a function of CT state energy in devices with the 14 NDI derivatives. The CT state energy is calculated by the peak wavelength of the CT emission in the EL spectra of the devices. We find a negative correlation; specifically, the TTA-UC emission intensity increases as the CT state energy decreases. Since the emission layers of all devices used the same 1,2-ADN, the difference in TTA-UC emission intensity was caused by the difference in energy transfer efficiency from $CT_3$ to $T_1$ of 1,2-ADN. The highest TTA-UC emission intensity was observed in the 1,2-ADN/NDI-HF device with a CT state energy of 1.71 eV (Fig. 3d), which is very close to the peak energy of phosphorescence (1.77 eV) from $T_1$ of 1,2-ADN in solution at 77 K (Fig. S28). Although the sample form and temperature of the measurement for the energy level are different, the result clearly indicates that the proximity of the energy levels between the CT state and $T_1$ accelerates the energy transfer between the two excitonic states. The triplet energy transfer is governed by the Dexter mechanism[29]. Therefore, a larger spectral overlap between the two states will facilitate energy transfer[30,31]. We suppose that the bulkiness of the side chain of NDI derivatives is another possible factor determining the energy transfer efficiency because the top three derivatives (NDI-HF, NDI-Ph, NDI-FPh), which show strong TTA-UC emission, have small steric hindrance to the π-plane of the NDI core. In particular, NDI-HF has such strong intermolecular interactions that it cannot be dissolved in any organic solvent. The strong intermolecular interaction is beneficial for a large heteromolecular CT interaction at the interface. Our finding is further supported by UC-OLEDs using another type of anthracene derivative TPA-An-mPhCz[32] with a 0.2 eV shallower HOMO level than 1,2-ADN and with a $T_1$ level almost identical to that of 1,2-ADN (Fig. S29). Corresponding to the shallower HOMO level, the CT state

energies of the device with TPA-An-mPhCz are 0.2–0.3 eV smaller than those of 1,2-ADN when the same NDI is used, as shown in Fig. S30. Notably, the difference in the CT state energy changes the optimal acceptor partner for efficient TTA-UC emission in TPA-An-mPhCz. NDI-HF is optimal for 1,2-ADN, whereas NDI-Cy with a shallower LUMO level relative to NDI-HF is optimal for TPA-An-mPhCz, as shown in the plot of the TTA-UC emission intensity of the devices (Fig. S30d). The CT state energy of the TPA-An-mPhCz/NDI-HF device is as low as 1.40 eV; therefore, the energy transfer from the CT state to $T_1$ of the emitter is suppressed. With the optimal combination, the TPA-An-mPhCz/NDI-Cy device, which has close energy levels between the CT state and $T_1$ of the emitter, shows an ultralow turn-on voltage, similar to that observed for 1,2-ADN/NDI-HF (Fig. S31a). Notably, B4PYMPM, which is a typical electron transport material with a deep LUMO level, does not function at a low turn-on voltage even with TPA-An-mPhCz, which has a shallower HOMO level. CT state absorption (Fig. S31b) was observed in the TPA-An-mPhCz/NDI-Cy device but not in the TPA-An-mPhCz/B4PYMPM device. This result further supports our conclusion that the CT state does not form on materials with weak intermolecular interactions even when the energy difference between the HOMO of the donor and LUMO of the acceptor is small, and CT state formation at the D/A interface is essential for direct excitation of $T_1$ at an ultralow voltage.

Thus far, the best D/A combination showing efficient TTA-UC emission is 1,2-ADN/NDI-HF. We optimized the device structure by adding a typical blue fluorescent dopant, *tert*-butyl perylene (TbPe)[33], in the emitter layer. The device structure is illustrated in Fig. 4a. The TbPe-doped layer is sandwiched by the undoped 1,2-ADN layers. The device concept is as follows: the core processes, including CT state formation, energy transfer to $T_1$, and TTA, occur near the D/A interface;

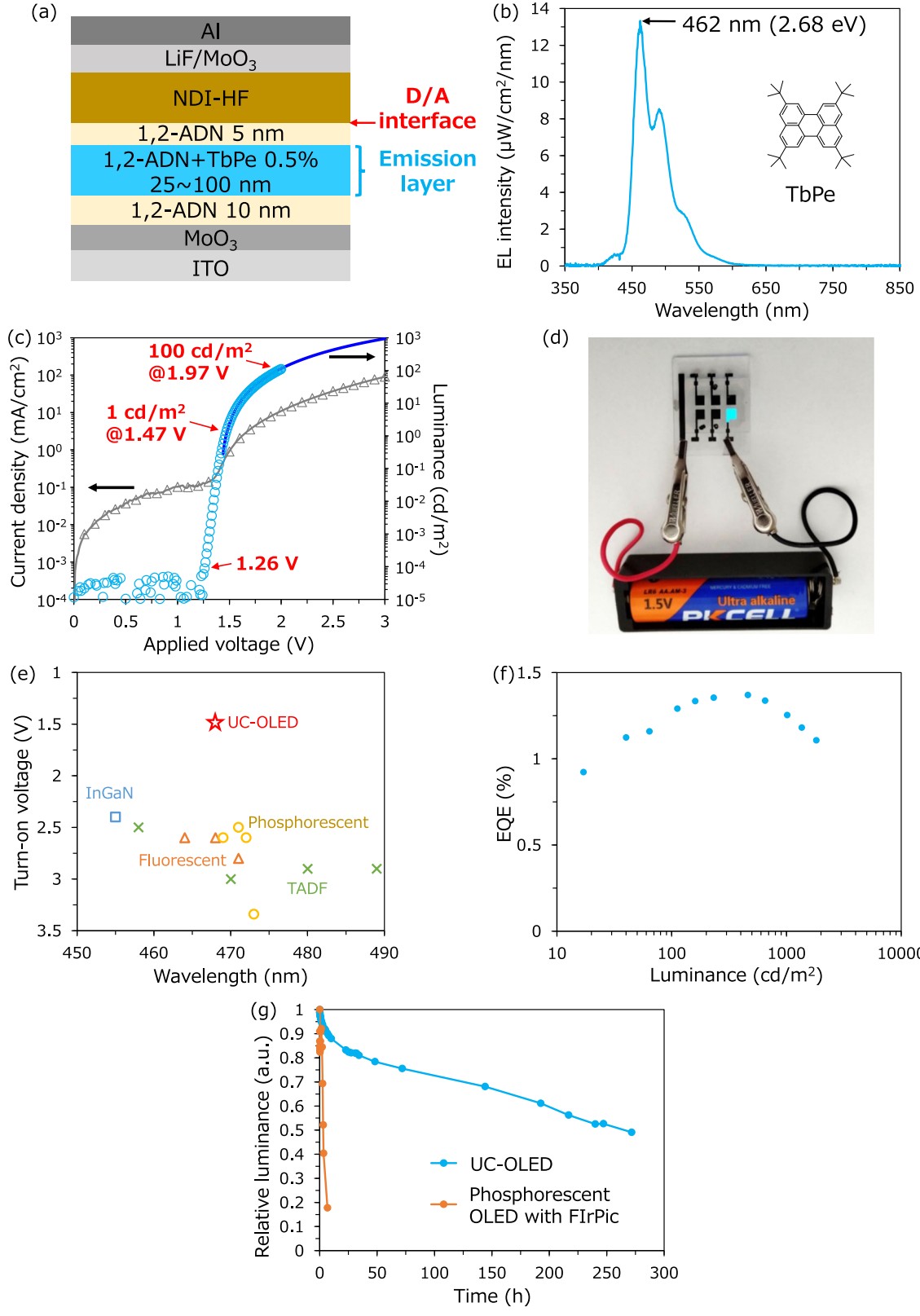

**Fig. 4 | Optimized device performances. a** Schematic of the optimized device structure with TbPe as a fluorescent dopant. **b** EL emission spectrum of the TbPe-doped 1,2-ADN/NDI-HF device under a constant current flow (100 mA/cm²). The inset shows the chemical structure of TbPe. **c** *J*−*V* (gray triangle) and *L*−*V* curves for the TbPe-doped device. The blue curve is measured by a luminance meter, and the sky-blue circle is measured by the photodiode. The value of the photodiode is corrected by multiplying by a coefficient to obtain the same value measured by the luminance meter. **d** Photograph of a TbPe-doped 1,2-ADN/NDI-HF device operated by a 1.5 V battery. **e** Turn-on voltage versus EL emission wavelength of various types of blue LEDs: UC-OLED (red star), fluorescent OLED (orange triangle), phosphorescent OLED (yellow circle), TADF OLED (green cross) and InGaN LED (blue square). The device parameters for the references used to create the figure are summarized in Table S2. **f** EQE of the TbPe-doped 1,2-ADN/NDI-HF devices. **g** The operation lifetime measurement under the initial luminance condition at 1000 cd/m² of the TbPe-doped UC-OLED device (blue) and the FIrPic-doped phosphorescent device (orange).

however, the final emission occurs apart from the interface by Förster energy transfer to TbPe to suppress interfacial quenching of the emissive $S_1$ exciton[14]. The EL emission spectrum and $J$-$V$ and $L$-$V$ curves of the optimized device are depicted in Fig. 4b, c. The EL spectra at different current densities are shown in Fig. S32. The device produced blue emission with a peak wavelength at 462 nm (2.68 eV) from TbPe. The emission from other species, such as the CT state and $T_1$ appearing from 650 ~ 850 nm, is less than the noise level (intensity less than 0.3% of the main emission from TbPe). Notably, the turn-on voltage that shows 1 cd/m² was only 1.47 V, producing 100 cd/m² at 1.97 V. The high-energy blue emission is still observable at 1.26 V when sensitively measured by a photodiode. The blue emission of the UC-OLED is also visible by only connecting a 1.5 V battery, as shown in Fig. 4d. The turn-on voltage versus EL emission wavelength of various types of blue OLEDs and an inorganic blue LED based on InGaN is plotted in Fig. 4e. The parameters of the reference devices are summarized in Table S2. Recently, several papers have been published reporting low-voltage operation of blue OLEDs[34,35]. However, the turn-on voltage was approximately 2.5 V. The ultralow voltage operation at approximately 1.5 V for blue emission has not been achieved even with inorganic LEDs[36]. Therefore, this is the lowest operating voltage thus far among any type of blue LED. Some of the OLEDs in Table S2 exhibit EQE values greater than 20% with a low turn-on voltage of ~2.5 V. These devices use either phosphorescent or TADF materials, which have not been utilized in commercial blue OLEDs due to their low stability. Therefore, reducing the operating voltage of blue OLEDs using fluorescent materials is crucial. The $L$-$J$ curve of the TbPe-doped device has been added to Fig. S33. Luminance increases quadratically at lower current density and linearly at higher current density. This is a well-known characteristic of TTA, namely, bimolecular TTA becomes a main decay channel under high triplet concentration conditions, and TTA efficiency becomes constant[37]. The transient EL of the TbPe-doped device under different applied voltages is shown in Fig. S34. Only a slow decay component with a lifetime on the order of µs is observed, similar to the result of the undoped device in Fig. 2f, indicating that all the emission is produced from the low-energy $T_1$ of 1,2-ADN even in the doped device regardless of the applied voltage. The maximum EQE of the UC-OLED is 1.37% (Fig. 4f). The EQE of the UC-OLED was largely improved with the fluorescent dopant because parasitic loss processes such as back charge separation were suppressed, as previously reported[14]. The power efficiency (PE) is 3.04 lm/W (Fig. S35). The EQE and PE are comparable to those of conventional blue fluorescent OLEDs[3,38]. The theoretical maximum EQE for the UC-OLED is calculated by multiplying the efficiency for every step used to produce blue emission; the spin-statistics of triplet formation are 75%[39], the maximum TTA efficiency is 50% because TTA is a two-photon process[40], the measured photoluminescence QE of the TbPe-doped 1,2-ADN film is 73%, and the outcoupling efficiency is 20%[41,42]. Therefore, the theoretical maximum EQE for the blue UC-OLED is calculated to be 5.5%. Here, we assume that only the triplet CT state is used for TTA-UC emission; however, the singlet CT state is possibly used because the singlet and triplet CT states are almost degenerate, leading to fast intersystem crossing[39,43,44]. If both singlet and triplet CT states are used, the theoretical maximum EQE is calculated to be 7.3%. The difference between the theoretical maximum EQE and current value is possibly caused by the deactivation processes, such as direct CT recombination before the energy transfer from the CT state to $T_1$ of the emitter or triplet-charge annihilation. Suppression of these processes by finding new material and optimizing the device structure will further increase the EQE of the UC-OLED. Note that the number of excitons is halved through TTA-UC. However, 37.5% corresponds to half of the formation ratio of $T_1$ (75%/2) and is still larger than 25%, which is the formation ratio of $S_1$ utilized in conventional fluorescent OLEDs[2]. Efficiency roll-off was observed at operating conditions above 1000 cd/m², as shown in Fig. 4f, and this is probably caused by triplet-charge annihilation[14,45].

As the sky-blue emission from TbPe is not appropriate for display applications, we tried a famous narrow-band blue emitter, ν-DABNA, as a fluorescent dopant in the 1,2-ADN layer[46]. Pure blue emission from ν-DABNA was observed; however, the turn-on voltage increased to 2.7 V due to an increase in the series resistance of the device, as shown in Fig. S36. What kind of dopant materials can be appropriate in UC-OLEDs will be investigated in the future. The operation lifetime of the blue UC-OLED is compared to the device with a typical blue phosphorescent emitter, the bis-cyclometalated iridium (III) complex (FIrPic)[47,48], under the initial luminance condition at 1000 cd/m² (Fig. 4g). The devices were simply encapsulated by UV resin and a cover glass without using desiccant and an oxygen scavenger. FIrPic exhibited phosphorescence from $T_1$ at a peak wavelength of 475 nm (2.61 eV), as shown in Fig. S37. The lifetime for the luminance to decay to 50% from the initial luminance of 1000 cd/m² ($LT_{50}$) for the FIrPic device was found to be approximately 3 h due to device degradation by the high energy $T_1$. Conversely, the UC-OLED with a structure of Fig. 4a provided an $LT_{50}$ up to 270 h. This indicates that blue emission originating from low energy $T_1$ (1.77 eV in 1,2-ADN) is beneficial for device stability. We also measured the operation lifetime of TbPe-doped devices with B4PYMPM or BPyOXD as the electron transport layer (Fig. S38). The $LT_{50}$ of the device with B4PYMPM or BPyOXD was 0.7 h and 2.5 h, respectively; these values are much shorter than that of the UC-OLED with NDI-HF. The short lifetime of the TbPe-doped device using the conventional electron transport layer is probably due to exciton quenching at the $MoO_3$ interface. Durable OLEDs generally use a multilayered structure with electron, hole, and exciton blocking layers and a strategy for shifting the emission zone[49]. The device structure of the UC-OLED is a simple D/A heterojunction, but it shows relatively high stability. Notably, the UC-OLED worked even without the LiF electron injection layer, as shown in Fig. S39. This result was caused by a reduced energy barrier for electron injection due to the much lower LUMO level of NDI-HF than that of 1,2-ADN. An electron injection material such as LiF is one of the main causes of the reduced operational lifetime of OLEDs because of its air sensitivity[50]. Thus, the present UC-OLED system could have further enhanced the operation lifetime by avoiding the use of problematic elements.

## Discussion

Finally, we discuss the origin of the extremely small starting voltage of the blue emission at 1.26 V, as shown in Fig. 4c. To investigate the dependence of the applied electric field, the thickness dependence of the emission layer on the emissive properties was investigated. As shown in Fig. S40, the threshold of the $L$–$V$ curves was not shifted by the difference in the emission layer thickness between 25 and 100 nm. Specifically, the threshold voltage for the blue emission was not influenced by the electric field, indicating that the emissive properties were determined essentially by the D/A interface where the CT state formed. Note that 1.26 V, the threshold voltage for obtaining blue emission (Fig. 4c), was much smaller than the CT state energy of 1,2-ADN/NDI-HF (1.71 eV) divided by the elementary charge. A recent report demonstrated that the EL emission in any type of inorganic and organic LED was observable by highly sensitive photon counting measurements at 0.5–1.0 V smaller than the bandgap energy of the emitter divided by the elementary charge[36]. The authors explained that the origin of the emission was the radiative recombination of nonthermal-equilibrium band-edge carriers whose populations were determined by the Fermi-Dirac function perturbed by a small external bias. In the case of organic semiconductors, band-edge carriers exist in the tail states inside the band gap of the material[51]. Those states determined the diode characteristic of the devices, especially near the threshold. The diode characteristics of our UC-OLED are shown in Fig. S41. The threshold voltage of the diode for the current flow is at ~1.3 V, and the open-circuit voltage ($V_{OC}$) of the UC-OLED under 1 sun

irradiation is 1.32 V. These values correlate to an ultralow threshold voltage for blue light-emissive OLEDs.

In summary, we have demonstrated that the UC-OLED has an ultralow turn-on voltage of 1.47 V for emitting blue light with a peak wavelength at 462 nm (2.68 eV) and reaches 100 cd/m$^2$, which is equivalent to the luminance of a typical display, at 1.97 V. Blue emission is achieved by the selective excitation of low-energy $T_1$ at low applied voltage and TTA-UC emission near the D/A interface. The essential factor is the appropriate choice of the D/A material to achieve CT state formation and subsequent efficient energy transfer to $T_1$ of the emitter at the D/A interface. We have discovered that the formation of the CT state is influenced by the strong intermolecular interaction between the donor and acceptor materials. The importance of our findings lies in the reduction of the turn-on voltage using the emitter commonly utilized in commercial OLEDs. Our findings have a great impact on advancements in the field not only for OLEDs but also for OPVs because $T_1$ formation at the D/A interface is currently considered the main cause of nonradiative recombination; this is the last challenging topic of OPVs for achieving a power conversion efficiency of over 20%[52]. We believe that the appropriate design of the D/A interface is essential for controlling the dynamics of excitonic processes[53], leading to the development of efficient organic electronic devices and novel optoelectronic functions.

## Methods

### OLED fabrication

The OLED devices were fabricated on indium tin oxide (ITO)-coated glass substrates (ITO thickness: 150 nm; sheet resistance: 10.3 Ω sq$^{-1}$; Techno Print). 9-(Naphthalen-1-yl)-10-(naphthalen-2-yl)anthracene (1,2-ADN) (sublimed, Lumtec), 4-(10-(3-(9H-carbazol-9-yl)phenyl) anthracen-9-yl)-N,N-diphenylaniline (TPA-An-mPhCz) (sublimed, Lumtec), 2,7-di(9H-fluoren-2-yl)benzo[lmn][3,8]-phenanthroline-1,3,6,8 (2H,7H)-tetraone (NDI-HF) (sublimed, Lumtec), 2,7-diphenylbenzo [lmn][3,8]phenanthroline-1,3,6,8(2H,7H)-tetraone (NDI-Ph) (sublimed, Lumtec), 2,7-dioctylbenzo[lmn][3,8]phenanthroline-1,3,6,8(2H, 7H)-tetraone (NDI-C8) (sublimed, Lumtec), 2,7-dicyclohexylbenzo[lmn] [3,8]phenanthroline-1,3,6,8(2H,7H)-tetraone (NDI-Cy) (sublimed, Lumtec), 4,6-bis(3,5-di(pyridin-4-yl)phenyl)-2-methylpyrimidine (B4PY MPM) (sublimed, Lumtec), 1,3-Bis[2-(2,2′-bipyridine-6-yl)-1,3,4-oxa-diazo-5-yl]benzene (BPyOXD) (sublimed, Lumtec), 1,3,5-tri[(3-pyridyl)-phen-3-yl]benzene (TmPyPB) (sublimed, Lumtec), and 2,5,8,11-tetra-tert-butylperylene (TbPe) (Tokyo Chemical Industry) were used without further purification. The MoO$_3$ hole-transporting layer (10 nm, 0.01 nm s$^{-1}$), donor layer (40-115 nm, 0.1 nm s$^{-1}$), acceptor layer (50 nm, 0.1 nm s$^{-1}$), LiF electron-injection layer (0.2 nm, 0.01 nm s$^{-1}$), MoO$_3$ layer (0.3 nm, 0.01 nm s$^{-1}$) and Al electrodes (70 nm, 0.1 nm s$^{-1}$) were deposited by thermal evaporation under high vacuum (-10$^{-5}$ Pa) in a vacuum evaporation system (VTS-350M, ULVAC) housed in a glove box (DSO-1.5 S MS3-P, Miwa). TbPe were introduced into the 1,2-ADN donor layer via the codeposition technique. The TbPe concentration are described as the volume percent relative to the 1,2-ADN volume in the film. The devices were simply encapsulated in the glove box by UV-resin and a cover glass without using desiccant and an oxygen sca-venger. The active area of the devices was 0.065 cm$^2$.

### Measurements

The J–V and L–V curves of the OLEDs were measured using a source/measure unit (2400, Keithley) and a luminance meter (LS-110, KONICA MINOLTA, INC.) or a source/measure unit (B2902A, Keysight Technologies Inc.) and a luminance meter (BM-9, TOPCON Ltd.). The highly sensitive L–V curves were measured by a photodiode (S1223-01, Hamamatsu Photonics) and a semiconductor parameter analyzer (B1500A, Keysight Technologies Inc.) combined with a high-resolution source/measure unit (B1517A HRSMU, Keysight Technologies Inc.). The EL intensity measured by the photodiode was corrected by multiplying

a coefficient so as to be the same value measured by the luminance meter. A fiber optic spectrometer (Avaspec-UV/VIS/NIR, Avantes) was used to measure the absolute EL spectra of the OLEDs. Absolute EL spectra were measured when a constant current was applied (2400, Keithley) or (B2902A, Keysight Technologies Inc.); these spectra were collected by a cosine collector (CC-VIS/NIR, Avantes) attached to the end of an optical fiber to cover all the emission region of the device. The external quantum efficiency (EQE) was calculated between integrating the absolute EL spectrum and the measured current value. EL decay was measured by a hand-made system composed of a digital oscilloscope (T3DSO1302A, Teledyne Lecroy), a Si-photodiode (FDS100, Thorlabs), and a transimpedance amplifier (AMP140, Thor-labs) when the function generator (WF1974, NF Corp.) was applied a square voltage of 100 Hz to the OLEDs. EL decay waveforms were measured using oversampling at a rate of 2 GS/s and smoothing by a moving average of 128 times.

The J–V characteristics of the photovoltaic device were measured under simulated solar illumination (AM 1.5, 100 mW cm$^{-2}$) from a solar simulator based on a 300 W Xe lamp (HAL-320, Asahi Spectra) using a source meter (2400, Keithley). The light intensity was calibrated with a standard silicon solar cell (CS-20, Asahi Spectra). The highly sensitive IPCE of the devices was measured on a double monochromator system (HQE-25DK, Bunkohkeiki) with sub-femtoamp source meter (6430, Keithley).

The absolute PLQE were measured by a spectrometer system, including a Xe lamp, monochromator, an integration sphere, and a multichannel detector (Quantaurus-QY, Hamamatsu Photonics). The film thicknesses were measured by using a surface profilometer (Dektak 150, Veeco). UV/vis and fluorescence spectra in the solution states were recorded on a spectrophotometer (V-630, JASCO) and a spectrofluorometer (FP-6200, JASCO), respectively. Redox potentials were measured by cyclic voltammetry (CV) on a potentiostat/galvanostat (HAB-151A, Hokuto Denko). The CV was measured in CH$_2$Cl$_2$ with a 0.1 M n-Bu$_4$NPF$_6$ as a supporting electrolyte. Glassy carbon, Pt wire, Ag/AgNO$_3$ were used as working electrodes, counter electrodes, and reference electrodes, respectively. The scan rate was 0.05 V/s. The ferrocene/ferrocenium ion redox couple was used as an external reference. The $^1$H and $^{13}$C NMR spectra were recorded on a JEOL JNM-AL300 spectrometer in CDCl$_3$ or DMSO-$d_6$ operating at the frequencies of 300 and 75 MHz, respectively. Chemical shifts are reported in parts per million (ppm) relative to CDCl$_3$ (77.16 ppm for $^{13}$C NMR), DMSO (2.50 ppm for $^1$H NMR and 39.5 ppm for $^{13}$C NMR), or tetramethylsilane (0.00 ppm for $^1$H NMR) as an internal standard in CDCl$_3$, and the coupling constants are reported in hertz (Hz). High-resolution electrospray ionization mass spectra (HR-ESI-MS) were measured for solution samples in CHCl$_3$/MeOH on a Waters Xevo Q-TOF UPLC-MS system with a positive ion mode. X-ray diffraction (XRD) patterns were measured with an X-ray dif-fractometer (Smart Lab, Rigaku) with Cu Kα at 45 kV and 200 mA. The photoelectron yield spectra (PYS) were measured using an (AC-2, Riken-Keiki). Phosphorescence spectra were measured with a spectrofluorometer (FP-8650, JASCO).

### Preparation of NDI derivatives

NDI-HF, NDI-Ph, NDI-C8, and NDI-Cy were purchased from Lumtec. NDI-Naph, NDI-PhE, NDI-EH, NDI-EP, and NDI-PH were synthesized from naphthalene-1,4,5,8-tetracarboxylic dianhydride according to literatures[54–56]. NDI-FPh, NDI-BPh, NDI-MeF, NDI-MePh, and NDI-TbPh were unprecedented compounds. Their synthesis and characterization data were described in the Supplementary Information.

## Data availability

The main data supporting the findings of this study are available within the article and its Supplementary Information. Extra data are available from the corresponding authors on request.

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

## Acknowledgements

This research was supported in part by JSPS KAKENHI, Grants-in-Aid for Scientific Research (19K04465, 20KK0323, 21H05411, 22K14592), JST PRESTO (JPMJPR2101), a project (JPNP20004) subsidized by the New Energy and Industrial Technology Development Organization (NEDO), Izumi Science and Technology Foundation, Shorai Foundation for Science and Technology, and The Morino Foundation for Molecular Science. The authors would like to thank K. Tajima and K. Nakano at RIKEN for their assistance with the X-ray diffraction and photoelectron yield measurements.

## Author contributions

S.I. conceived the idea, directed the project, fabricated the OLED devices, conducted the OLED characterization, and wrote most of the paper. M.M. conducted the OLED characterization, especially the transient EL and luminance measurements using the photodiode. K.F. and K.B. designed and synthesized NDI derivatives. Y.M., M.T., S.N., and M.H. supervised the research. All the authors reviewed the manuscript.

## Competing interests

The authors declare no competing interests.
