## [Peer Review File · Nature Communications]

Blue Organic Light-Emitting Diode with a Turn-on Voltage of 1.47 VREVIEWER COMMENTS

Reviewer #1 (Remarks to the Author):

This manuscript reports an ultralow voltage turn-on at 1.47 V for blue emission with a peak wavelength at 462 nm (2.68 eV) is demonstrated in a triplet-triplet up-conversion (TTU-UC) organic light-emitting device. Upon the rational selection of the donor (emitter) and acceptor (electron-transport) materials, excitons are formed at the donor/acceptor interface and subsequently transfer the low-lying T1 state of the emitter. More important, the OLED reaches 100 cd/m², which is equivalent to the luminance of a typical display, at 1.97 V. An in-depth explanation of the turn-on voltage and the operation mechanism regarding to the use of B4PYMPM and NDI-HF as the EML by using incident photon-to-current conversion efficiency (IPCE) to measure the CT state absorption, which reflects the D/A interaction at the interface between the emitter (donor) and the electron transport (acceptor) materials. The NDI-HF-based device showed a photocurrent response at around 700 nm further supporting the importance of the formation of the CT state as its serving as the intermedia state for energy transfer to T1 state of the emissive and eventually generates light. The Manuscript is good in quality with high impact to the OLED community. Several questions or concerns should be addressed before the publication in Nature Communication.

1. For figure 2(a), the transient EL spectra for BPyOXD and TmPyPB are low in quality. What is the reproducibility of the results? Any reasons that lead to weak intensity of the transient EL spectra? Are these attributed to the low photoluminescence quantum yields of the 1,2-ADN/BPyOXD and 1,2-ADN/TmPyPB layer? Please clarify.

2. Why NDI-HF derivatives are so special to generate CT state for TTU-UC mechanism, given that the energy difference of LUMO between B4PYMPM and NDI-HF is only 0.1 eV? What if the use of ET material with the same LUMO level to NDI-HF, could this CT state still be generated? Could it be possible the molecular stacking of the NDI with anthracene derivatives are also an important parameter? Please comment.

3. The TTA-UC emission intensity of the devices BPh is of similar to HF, given that they are having the same LUMO levels. Are these results suggest that the confinement of the LUMO level is one of the key parameters we have to take into consideration, but there are more and depth reasons or mechanisms we have to think of? Please comment.

4. The LUMO level of all NPD derivatives should be listed in table S1 to enhance the readability of the paper.

5. Some references should be added for the representation of the earlier works in phosphorescent and TADF materials, for example, 10.1038/s41566-022-00958-4; 10.1038/s41566-018-0332-z.

Reviewer #2 (Remarks to the Author):

In this work, Izawa and coworkers utilize charge transfer state mediated triplet triplet annihilation to develop blue OLEDs that work at very low voltages. They conclusively demonstrate the necessity of a CT state for the transfer mediation to work and extract the relationship between CT state energy and the efficiency of the mechanism. Finally, they utilize a doped upconversion layer to build a device capable of running at under two volts. Although TTA has been abundantly utilized in OLEDs in the past and this type of low-voltage turnon has been repeatedly demonstrated in rubrene-based devices, this is an important step forward for the technology due to the low turn-on voltages utilized for blue, and I am supportive of publication after the authors address the following concerns.

First, some of the data is internally inconsistent. For example, the J-V curves of the B4PYMPM and TmPbPB devices show strong jumps in current (fig S1) yet the L-V curves are completely smooth. What is happening here? Is the EQE really jumping around to make up for these discrepancies? I recommend plotting both data on the same plots. Similarly, the J-V should be included in Fig 4.

A likely primary deactivation channel in this device is triplet-charge annihilation. Do the authors see any sign of it? Is it responsible for the reduced efficiency relative to the optimum?

TTA is well known to be intensity dependent. A discussion of how this dependence compares to the carrier concentrations in the LED would be of interest.

A device structure would be useful for Figure 2.

There's a substantial redshift to sky blue for the TbPe device. Is it possible to design a higher energy system? The authors should also report the EQE for the devices in Fig 2, even if they are low.

Putting the stability curve in context with other blue materials would be helpful – FirPic is well known for having a particularly short lifetime among blue emitters. That particular device structure is clearly quite poor, and may be negatively affecting the degradation time.

Reviewer #3 (Remarks to the Author):

The manuscript submitted by Hiramoto et al. fabricate blue OLEDs with ultralow turn-on voltage of 1.47 V, which represents the lowest turn-on voltage among any blue LED. Moreover, the lifetime of device (T50) is 275 h, which beyond most blue TADF and phosphorescent devices. The authors consider that selective excitation of a low energy T1 in a conventional emitter and subsequent TTA-based fluorescence with high efficiency could effectively reduce the turn-on voltage. In order to prove the concept, fourteen NDI derivatives acceptor with different aryl or alkyl groups are synthesized. Meanwhile, large number of combinations between emitter (donor) and acceptor as interface to determine TTA-UC emission. Photophysical, electrochemical and electroluminescent properties of the emitters and devices are comprehensively investigated.

However, there are some key points unclear, making this paper can not be accepted at this stage. The authors need to further demonstrate the potential and mechanism of this kind of TTA based blue OLEDs.

1. Beside low operation voltages, the other performances, e.g. luminance, EQE and power efficiencies, of the devices were just ordinary and even low than most of pure fluorescence diodes (10000 nits, 3% EQE, T50 > 1000 for blue). I wonder is there any space for further performance improvement for this kind of devices?
2. Obviously, triplet excitons should be involved in the EL process, however, the efficiencies were still quite low. So, I wonder the exciton dynamics and key exciton quenching factors of this kind of devices.
3. The mechanism and materials design were still unclear. Especially, it seems NDI unit is the key; however, NDI derivatives are not popular as ETM. So, it is curious why NDI-HF became the best one. Can the electron affinity and mobility of the ETMs influence the device performances? Figure S1 showed that the IV characteristics of B4PYMPM and TmPyPB devices were not worse than the other devices, although B4PYMPM and TmPyPB are good ETMs. What is the reason?

Detailed comments:

1. Anthracene derivatives are modified on the 9, 10-position. Therefore, 1,2-ADN should replace by 9, 10-AND.
2. Taking wavelength and turn-on voltage as x-axis and y-axis, respectively, draw a graph to summarize literatures with turn-on voltage less than 2.5V.
3. Considering the energy transfer from CT state to T1 of the emitter, the estimated EQE should reevaluate.
4. Why the intensity of UC-OLED based on NDI-HF acceptor is maximum among fourteen NDI derivatives.
5. The lifetime of slow decay based on NDI-HF acceptor should be provided to assess the TTA and diffusion process.

The following are the original referees' comments (in italics) and our point-by-point responses. Document files with the revised parts highlighted in yellow.

Reviewer #1 (Remarks to the Author):

This manuscript reports an ultralow voltage turn-on at 1.47 V for blue emission with a peak wavelength at 462 nm (2.68 eV) is demonstrated in a triplet-triplet up-conversion (TTU-UC) organic light-emitting device. Upon the rational selection of the donor (emitter) and acceptor (electron-transport) materials, excitons are formed at the donor/acceptor interface and subsequently transfer the low-lying T1 state of the emitter. More important, the OLED reaches 100 cd/m², which is equivalent to the luminance of a typical display, at 1.97 V. An in-depth explanation of the turn-on voltage and the operation mechanism regarding to the use B4PYMPM and NDI-HF as the EML by using incident photon-to-current conversion efficiency (IPCE) to measure the CT state absorption, which reflects the D/A interaction at the interface between the emitter (donor) and the electron transport (acceptor) materials. The NDI-HF-based device showed a photocurrent response at around 700 nm further supporting the importance of the formation of the CT state as its severing as the intermedia state for energy transfer to T1 state of the emissive and eventually generates light. The Manuscript is good in quality with high impact to the OLED communality. Several questions or concerns should be address before the publication in Nature Communication.

1. For figure 2(a), the transient EL spectra for BPyOXD and TmPyPB are low in quality. What is the producibility of the results? Any reasons that lead to weak intensity of the transient EL spectra? Are these attributed to the low photoluminescence quantum yields of the 1,2-ADN/BPyOXD and 1,2-ADN/TmPyPB layer? Please clarify.

Response: We thank the reviewer for the valuable comments and high-quality evaluation of our manuscript. We made the devices again and measured the transient EL spectra. All the spectra have small noise levels because the device quality improved. Figure 2f has been replaced.

2. Why NDI-HF derivatives are so special to generate CT state for TTU-UC mechanism, given that the energy difference of LUMO between B4PYMPM and NDI-HF is only 0.1 eV? What if the use of ET material with the same LUMO level to NDI-HF, could this CT state can still be generated? Could it be possible the molecular stacking of the NDI with anthracene derivatives are also an important parameter? Please comment.

Response: Thank you for the valuable comment. We have added the XRD patterns of B4PYMPM, BPyOXD, TmPyPB, and NDI-HF in Figure S3. There are no clear diffraction

peaks in the XRD patterns of B4PYMPM, BPyOXD, and TmPyPB, indicating that these conventional electron transport materials are amorphous in thin films due to weak intermolecular interactions. In contrast, NDI-HF showed diffraction patterns, and the intermolecular interaction of NDI-HF is too strong to dissolve in any organic solvent. The strong intermolecular interaction of NDI-HF leads to a large heteromolecular interaction with 1,2-ADN at the interface; therefore, only NDI-HF forms a CT state at the interface. The discussion has been added in line 9, page 6. This hypothesis is further supported by the experiment using TPA-An-mPhCz, which has a 0.2 eV higher HOMO level than 1,2-ADN, as shown in Figure S31. B4PYMPM does not generate a CT state with TPA-An-mPhCz, and a TPA-An-mPhCz/B4PYMPM device does not show low-voltage turn on. This result indicates that the CT state does not form and that the UC-OLED does not work in materials with weak intermolecular interactions even when the energy difference between the HOMO of the donor and LUMO of the acceptor is small. This discussion has been added in line 12, page 8.

3. The TTA-UC emission intensity of the devices BPh is of similar to HF, given that they are having the same LUMO levels. Are this result suggest that the confinement of the LUMO level is one of the key parameters we have to take into consideration, but there are more and depth reasons or mechanisms we have to think of? Please comment.

Response: We suppose that the NDI-BPh mentioned by the reviewer is NDI-Ph because the TTA-UC emission intensity in the NDI-BPh device is ~100 times smaller than that in NDI-HF and ~50 times smaller than that in NDI-Ph. As we explained in Figure 3d, the proximity of the energy levels between the CT state and T_1 accelerates the energy transfer; therefore, the LUMO of NDI derivatives is very important because it determines the CT state energy. We suppose that the bulkiness of the side chain of NDI derivatives is another additional factor determining the energy transfer efficiency because Top 3 (HF, Ph, FPh), which shows strong TTA-UC emission, has small steric hindrance to the π -plane of the NDI core. In particular, NDI-HF has such strong intermolecular interactions that it cannot be dissolved in any organic solvent. The strong intermolecular interaction is beneficial for a large heteromolecular CT interaction at the interface. The discussion has been added in line 18, page 7.

4. The LUMO level of all NPD derivatives should be listed in table S1 to enhance the readability of the paper.

Response: We are not able to fully understand the meaning of the NPD derivatives the reviewer mentions; however, we suppose they are electron transport materials (B4PYMPM, BPyOXD, TmPyPB). The energy levels of the electron transport materials in the thin film were evaluated by photoelectron spectroscopy and absorption spectra in

previous studies, and those values are already listed in Figure 1c. The LUMO energy levels of NDI derivatives in solution were measured by CV and are listed in Table S1. The sample form and the measurement method are different; therefore, those values are listed in different figures and tables. The explanation has been added in line 16, page 4. To improve the readability of Table S1, we have changed the LUMO value in Table S1 from “relative to redox potential of ferrocene” to “relative to vacuum level” by assuming the redox potential of ferrocene relative to vacuum level is -4.8 eV.

5. Some references should be added for the representation of the earlier works in phosphorescent and TADF materials, for example, 10.1038/s41566-022-00958-4; 10.1038/s41566-018-0332-z.

Response: We thank the reviewer for the valuable suggestions. We have added two references on phosphorescent and TADF materials, including DOI: 10.1038/s41566-022-00958-4, and DOI: 10.1038/s41566-020-00745-z in line 4, page 3.

Reviewer #2 (Remarks to the Author):

In this work, Izawa and coworkers utilize charge transfer state mediated triplet triplet annihilation to develop blue OLEDs that work at very low voltages. They conclusively demonstrate the necessity of a CT state for the transfer mediation to work and extract the relationship between CT state energy and the efficiency of the mechanism. Finally, they utilize a doped upconversion layer to build a device capable of running at under two volts. Although TTA has been abundantly utilized in OLEDs in the past and this type of low-voltage turn on has been repeatedly demonstrated in rubrene-based devices, this is an important step forward for the technology due to the low turn-on voltages utilized for blue, and I am supportive of publication after the authors address the following concerns. First, some of the data is internally inconsistent. For example, the J-V curves of the B4PYMPM and TmPbPB devices show strong jumps in current (fig S1) yet the L-V curves are completely smooth. What is happening here? Is the EQE really jumping around to make up for these discrepancies? I recommend plotting both data on the same plots. Similarly, the J-V should be included in Fig 4.

Response: We thank the reviewer for the valuable comments and high-quality evaluation of our work. The jump in current mentioned by the reviewer was caused by the electrical instability of the devices. Therefore, we made the devices again and measured the J-V-L characteristics. The jump in current was not observed in the new devices, and those J-V curves are plotted in Figure 2d. The J-V and L-V curves of the doped device have been plotted together in Figure 4c following the reviewer's suggestion. However, the J-V and L-V curves of the four undoped devices are separated in Figure 2c and d because the figure becomes too busy.

A likely primary deactivation channel in this device is triplet-charge annihilation. Do the authors see any sign of it? Is it responsible for the reduced efficiency relative to the optimum?

Response: Triplet-charge annihilation is likely to occur at high charge concentration conditions. Efficiency roll-off was observed at operating conditions above 1000 cd/m² in Figure 4f, and this is probably caused by triplet-charge annihilation. The discussion has been added in line 27, page 9.

TTA is well known to be intensity dependent. A discussion of how this dependence compares to the carrier concentrations in the LED would be of interest.

Response: The L-J curve of the TbPe-doped device has been added to Figure S33. Luminance increases quadratically at lower current density and linearly at higher current

density. This is a well-known characteristic of TTA, namely, bimolecular TTA becomes a main decay channel under high triplet concentration conditions, and TTA efficiency becomes constant. The discussion has been added in line 5, page 9.

A device structure would be useful for Figure 2.

Response: We thank the reviewer for the valuable suggestion. We have added a device structure in Figure 2a.

There's a substantial redshift to sky blue for the TbPe device. Is it possible to design a higher energy system? The authors should also report the EQE for the devices in Fig 2, even if they are low.

Response: We have tried a famous narrow-band blue emitter, v-DABNA, as a fluorescent dopant in the 1,2-ADN layer. Pure blue emission from v-DABNA was observed; however, the turn-on voltage increased to 2.7 V due to an increase in the series resistance of the device, as shown in Figure S35. What kind of dopant material can be appropriate in UC-OLEDs will be investigated in the future. The discussion has been added in line 2, page 10. We have added the EQE of the undoped devices in Figure 2e following the reviewer's suggestion. The EQE of undoped UC-OLEDs is as low as 0.343%. The EQE was largely improved to 3.25% with the fluorescent dopant because parasitic loss processes such as back charge separation were suppressed, as we have previously reported in ref. 14. The discussion has been added in line 9, page 9.

Putting the stability curve in context with other blue materials would be helpful – FlrPic is well known for having a particularly short lifetime among blue emitters. That particular device structure is clearly quite poor, and may be negatively affecting the degradation time.

Response: We have also measured the operation lifetime of TbPe-doped devices with B4PYMPM or BPyOXD as the electron transport layer (Figure S37). The LT_{50} of the device with B4PYMPM or BPyOXD was 0.7 h and 2.5 h, respectively. The short lifetime of the TbPe-doped device using the conventional electron transport layer is probably due to exciton quenching at the MoO_3 interface. Durable OLEDs generally use a multilayered structure with electron, hole and exciton blocking layers and a strategy for shifting the emission zone. The device structure of the UC-OLED is a simple D/A heterojunction, but it shows relatively high stability. The discussion has been added in line 15, page 10.

Reviewer #3 (Remarks to the Author):

The manuscript submitted by Hiramoto et al. fabricate blue OLEDs with ultralow turn-on voltage of 1.47 V, which represents the lowest turn-on voltage among any blue LED. Moreover, the lifetime of device (T50) is 275 h, which beyond most blue TADF and phosphorescent devices. The authors consider that selective excitation of a low energy T1 in a conventional emitter and subsequent TTA-based fluorescence with high efficiency could effectively reduce the turn-on voltage. In order to prove the concept, fourteen NDI derivatives acceptor with different aryl or alkyl groups are synthesized. Meanwhile, large number of combinations between emitter (donor) and acceptor as interface to determine TTA-UC emission. Photophysical, electrochemical and electroluminescent properties of the emitters and devices are comprehensively investigated.

However, there are some key points unclear, making this paper can not be accepted at this stage. The authors need to further demonstrate the potential and mechanism of this kind of TTA based blue OLEDs.

1. Beside low operation voltages, the other performances, e.g. luminance, EQE and power efficiencies, of the devices were just ordinary and even low than most of pure fluorescence diodes (10000 nits, 3% EQE, T50 > 1000 for blue). I wonder is there any space for further performance improvement for this kind of devices?

Response: We thank the reviewer for the valuable comments. The theoretical maximum EQE for the blue UC-OLED is calculated to be 5.5%. We assume that only the triplet CT state is used for TTA-UC emission; however, the singlet CT state is possibly used because the singlet and triplet CT states are almost degenerate, leading to fast intersystem crossing. If both singlet and triplet CT states are used, the theoretical maximum EQE is calculated to be 7.3%. The current EQE of the device is 3.25%, and the origin of lagging behind the maximum is due to deactivation processes such as direct CT recombination or triplet-charge annihilation. Suppression of these processes further increases the EQE of the UC-OLED. The discussion has been added in line 17, page 9.

2. Obviously, triplet excitons should be involved in the EL process, however, the efficiencies were still quite low. So, I wonder the exciton dynamics and key exciton quenching factors of this kind of devices.

Response: TTA process halves the number of triplet excitons, and the calculated theoretical maximum EQE is 5.5%. The possible quenching mechanisms are direct CT recombination or triplet-charge annihilation. Suppression of these processes further increases the EQE of the UC-OLED. The discussion has been added in line 21, page 9.

3. The mechanism and materials design were still unclear. Especially, it seems NDI unit is the key; however, NDI derivatives are not popular as ETM. So, it is curious why NDI-HF became the best one. Can the electron affinity and mobility of the ETMs influence the device performances? Figure S1 showed that the IV characteristics of B4PYMPM and TmPyPB devices were not worse than the other devices, although B4PYMPM and TmPyPB are good ETMs. What is the reason?

Response: Electrical instability of the devices caused a jump in current of B4PYMPM and TmPyPB devices in previous *J-V* curves. Therefore, we made the devices again and measured *J-V-L* characteristics. The jump in current was not observed in the new devices, and those *J-V* curves are plotted in Figure 2d. NDI-HF is the best material for realizing efficient TTA-UC emission because it can generate a CT state at the interface. The next question is why NDI-HF is a special form of the CT state. We measured the XRD patterns of B4PYMPM, BPyOXD, TmPyPB, and NDI-HF in Figure S3. There are no clear diffraction peaks in the XRD patterns, indicating that these conventional electron transport materials are amorphous in thin films due to weak intermolecular interactions. In contrast, diffraction patterns are observed, and the intermolecular interaction of NDI-HF is too strong to be dissolved in any organic solvent. The difference in the intermolecular interaction determines whether the materials form a CT state with the anthracene derivative at the interface. The discussion has been added in line 9, page 6. This hypothesis is further supported by the experiment using TPA-An-mPhCz, which has a 0.2 eV higher HOMO level than 1,2-ADN, as shown in Figure S29a. B4PYMPM does not generate a CT state with TPA-An-mPhCz, and a TPA-An-mPhCz/B4PYMPM device does not show low-voltage turn on. This result indicates that the CT state does not form and that the UC-OLED does not work in materials with weak intermolecular interactions even when the energy difference between the HOMO of the donor and LUMO of the acceptor is small. This discussion has been added in line 12, page 8.

Detailed comments:

1. Anthracene derivatives are modified on the 9, 10-position. Therefore, 1,2-ADN should replace by 9, 10-ADN.

Response: The IUPAC name of 1,2-ADN is 9-(naphthalen-1-yl)-10-(naphthalen-2-yl)anthracene. "1,2-" means the substitution position of naphthalene, and it is a common abbreviation in this field, as used in refs. 17 and 37. We have added the IUPAC name of 1,2-ADN in Line 7, page 4.

2. Taking wavelength and turn-on voltage as x-axis and y-axis, respectively, draw a graph to summarize literatures with turn-on voltage less than 2.5V.

Response: We thank the reviewer for the valuable comments. We have added the plot of the turn-on voltage versus wavelength of EL emission in various types of blue LEDs in Figure 4e and added the explanation in line 27, page 8.

3. Considering the energy transfer from CT state to T₁ of the emitter, the estimated EQE should reevaluate.

Response: The estimated EQE was calculated by multiplying the triplet formation ratio (75%), maximum TTA efficiency (50%), measured photoluminescence quantum yield (73%), and outcoupling efficiency (20%). The efficiency of the energy transfer from the CT state to T₁ of the emitter cannot be measured directly; however, the difference between the theoretical maximum EQE and current value is possibly caused by a reduction in the energy transfer efficiency of these two states. Direct CT recombination is one possibility to reduce the energy transfer efficiency from the CT state to T₁ of the emitter. The discussion has been added in line 21, page 9.

4. Why the intensity of UC-OLED based on NDI-HF acceptor is maximum among fourteen NDI derivatives.

Response: As we explained in line 15, page 7, the proximity of the energy levels between the CT state and T₁ accelerates the energy transfer. The LUMO of NDI derivatives determines the CT state energy; therefore, the LUMO of NDI-HF is at the best position for realizing efficient TTA-UC emission. We suppose that the bulkiness of the side chain of NDI derivatives is another additional factor determining the energy transfer efficiency because Top 3 (HF, Ph, FPh), which shows strong TTA-UC emission, has small steric hindrance to the π -plane of the NDI core. In particular, NDI-HF has such strong intermolecular interactions that it cannot be dissolved in any organic solvent. The discussion has been added in line 18, page 7.

5. The lifetime of slow decay based on NDI-HF acceptor should be provided to assess the TTA and diffusion process.

Response: The decay curve of the 1,2-ADN/NDI-HF device has been fitted by a double exponential curve in Figure S2. The results reveal that the lifetime of TTA emission in 1,2-ADN/NDI-HF is 0.85 μ s, which is faster than the value in the UC-OLED with rubrene in a previous report (>3 μ s), indicating faster triplet diffusion in 1,2-ADN than in rubrene. The discussion has been added in line 16, page 5.

REVIEWER COMMENTS

Reviewer #1 (Remarks to the Author):

I have carefully reviewed the corrections made by the authors, and I think the authors have updated the figures and tables in the manuscript and supporting information respectively. As the authors have also pointed out the importance of the molecular packing for generating exciplex for high TTA-UC emission intensity confirmed as by their XRD studies, I would like to suggest that the discovery should also be mentioned in the conclusion part in the manuscript.

Reviewer #2 (Remarks to the Author):

My concerns are adequately addressed and I support publication.

Reviewer #3 (Remarks to the Author):

After carefully assessing the revised manuscript, authors have responded each comment according to reviewers' suggestions. Although all the spectra have low noise levels and the jump in current was not observed in new devices, no further performance improvement are achieved in new devices. Moreover, the low turn-on voltage of 1.47 V is at the sacrifice of EQE of devices (less than 1%). At the same time, the luminance of devices is below 10^3 cd m⁻², which are far from commercial criterion. So far, several blue OLEDs exhibit high efficiency and low voltage, such as Chem. Eng. J., 2023, 465, 142848. (Von: 2.8 V and EQE: 29.2%); Adv. Mater., 2023, 35, 2210413. (Von: 2.67 V and EQE: 28%); Angew. Chem. Int. Ed., 2022, e202205380. (Von: 2.8 V and EQE: 29%). In light of above viewpoints, there exists fatal problems in this manuscript, which can't match the profundity of Nature Communications. Therefore, I think it is not suitable to be published after revision.

1. As concerned by review III, triplet excitons should be involved in the EL process. However, the authors do not perform the exciton dynamics and key exciton quenching factors. Therefore, the EL time resolved emission spectra (TRES) should be added in this paper.

2. The maximum EQE based on TbPe-doped 1,2-ADN/NDI-HF upconversion-OLED is as low as 3.25%, which is far below the theoretical EQE of TTA upconversion-OLED (>10%). What are the advantages and highlights of this work?

3. For traditional fluorescence device, the luminance lower than 10^4 cd m⁻² is unacceptable. In this work, the luminance of TbPe-doped 1,2-ADN/NDI-HF device is at the scale of 10^3 cd m⁻², which is much lower than the level of TbPe-doped devices. Such as ACS Appl. Mater. Interfaces, 2019, 11, 26. (EQE: 15.3%); J. Mater. Chem. C, 2015, 3, 8834. (EQE: 18.1%).

4. There is no sense talking about so-called low turn-on voltage without emphasizing on the efficiency of device. Low turn-on voltage does not represent state-of-the-art device performance. It can also be achieved by external circuit engineering and device structure optimization. Several blue OLEDs exhibit high efficiency along with low turn-on voltage, such as Chem. Eng. J., 2023, 465, 142848. (Von: 2.8 V and EQE: 29.2%); Chem. Eur. J., (Von: 2.9 V and EQE: 20.6%); J. Mater. Chem. C, 2018, 6, 7839. (Von: 2.6 V and EQE: 26.4%). The crucial issue in this work is the unacceptable EQE of devices (< 3.52%), which is not match with the high-quality requirements of Nature Communications. Besides, authors are suggested to revised the title focusing on the scientific issue.

5. The authors do not carefully compare analogous literatures with low turn-on voltage. To our knowledge, the turn-on voltages of phosphorescent and TADF devices are less than 3.0 V, respectively. For instance, J. Mater. Chem. C, 2022, 10, 8349. (Von: 2.6 V); Dyes Pigm., 2022, 200, 110041. (Von: 2.4 V); J. Mater. Chem. C, 2021, 9, 10334. (Von: 2.6 V); J. Mater. Chem. C, 2021, 9, 260. (Von: 2.6 V). Moreover, authors should summarize the representative blue electroluminescence devices with turn-on voltages less than 3.0 V and list a table including device structure, turn-on voltages, luminance, efficiency, CIE and citation in Supplementary Information.

The following are the original reviewers' comments (in italics) and our point-by-point responses. Document files with the revised parts highlighted in yellow.

Reviewer #1 (Remarks to the Author):

I have carefully reviewed the corrections made by the authors, and I think the authors have updated the figures and tables in the manuscript and supporting information respectively. As the authors have also pointed out the importance of the molecular packing for generating exciplex for high TTA-UC emission intensity confirmed as by their XRD studies, I would like to suggest that the discovery should also be mentioned in the conclusion part in the manuscript.

Response: We thank the reviewer for the valuable comments and high-quality evaluation of our work. Following the reviewer's suggestion, a sentence mentioning that "We have discovered that the formation of the CT state is influenced by the strong intermolecular interaction between the donor and acceptor materials." has been added to the conclusion part of the manuscript.

Reviewer #2 (Remarks to the Author):

My concerns are adequately addressed and I support publication.

Response: We thank the reviewer for the high-quality evaluation of our work.

Reviewer #3 (Remarks to the Author):

After carefully assessing the revised manuscript, authors have responded each comment according to reviewers' suggestions. Although all the spectra have low noise levels and the jump in current was not observed in new devices, no further performance improvement are achieved in new devices. Moreover, the low turn-on voltage of 1.47 V is at the sacrifice of EQE of devices (less than 1%). At the same time, the luminance of devices is below 10^3 cd m⁻², which are far from commercial criterion. So far, several blue OLEDs exhibit high efficiency and low voltage, such as Chem. Eng. J., 2023, 465, 142848. (Von: 2.8 V and EQE: 29.2%); Adv. Mater., 2023, 35, 2210413. (Von: 2.67 V and EQE: 28%); Angew. Chem. Int. Ed., 2022, e202205380. (Von: 2.8 V and EQE: 29%). In light of above viewpoints, there exists fatal problems in this manuscript, which can't match the profundity of Nature Communications. Therefore, I think it is not suitable to be published after revision.

Response: We thank the reviewer for the valuable comments. As we had already reported in ref. 14 and discussed on page 9, line 22, the EQE of a UC-OLED without a fluorescent dopant becomes low because of parasitic loss processes such as singlet fission and back charge separation. However, the EQE of the blue UC-OLED with the fluorescent dopant was largely improved to 3.25%, from 0.34% in the undoped device. The doped UC-OLED reaches 1000 cd/m², as shown in Figure 4c.

The most important point we want to emphasize is that the references showing a high EQE that the reviewer raised are for blue OLEDs using either TADF or phosphorescent materials. As we discussed in the introduction, these materials have the intrinsic problem of low stability because the high-energy T1 causes their degradation, although they show high EQE. Therefore, conventional fluorescent materials are still used in commercial blue OLEDs. The importance of our findings lies in the reduction of the turn-on voltage using the emitter commonly utilized in commercial OLEDs. A discussion has been added to the abstract, introduction, and conclusion.

Another point is that the turn-on voltages in the references that the reviewer raised are 2.5~2.9 V, and a turn-on voltage of less than 1.5 V for blue emission has never been achieved. This is because the turn-on voltage is determined by the energy of the excited state formed after charge recombination. We have demonstrated a turn-on voltage of less than 1.5 V for the first time by passing through novel energy transfer steps for blue emission by using the interfacial charge transfer state as an intermediate.

1. As concerned by review III, triplet excitons should be involved in the EL process. However, the authors do not perform the exciton dynamics and key exciton quenching factors. Therefore, the EL time resolved emission spectra (TRES) should be added in

this paper.

Response: We already discussed the emission dynamics of undoped UC-OLEDs based on transient EL measurements in Figures 2f and S2. This is a common method to discuss the dynamics of the TTA in OLEDs. Additionally, we have newly added the result of the transient EL of the UC-OLED with TbPe as a fluorescent dopant under different applied voltages in Figure S34. Only a slow decay component with a lifetime on the order of μs is observed, similar to the result of the undoped device in Figure 2f, indicating that all the emission from the UC-OLED with TbPe is produced by the low-energy T1 of 1,2-ADN regardless of the applied voltage. A discussion has been added on page 9, line 18. Furthermore, we have added emission spectra of the UC-OLED with TbPe on a logarithmic scale in Figure S32b. Currently, we observe transient EL by a photodiode, as mentioned on page S3, line 19 in the Supplementary Information; therefore, we cannot measure the dynamics of the emission spectra. However, almost all the emission from the UC-OLED comes from the fluorescence of TbPe, and emission from other species, such as the CT state and T1 appearing from 650~850 nm, is less than the noise level (intensity less than 0.3% of the main emission from TbPe), as shown in Figure S32b. The discussion has been added on page 9, line 1. Therefore, the transient EL spectra would be expected to only show the strong fluorescence from TbPe, and the signal from the CT state and T1 would be expected to be buried in it. Therefore, we conclude that the transient EL in Figures 2f, S2, and S34 is sufficient to discuss the dynamics.

2. The maximum EQE based on TbPe-doped 1,2-ADN/NDI-HF upconversion-OLED is as low as 3.25%, which is far below the theoretical EQE of TTA upconversion-OLED (>10%). What are the advantages and highlights of this work?

Response: First, we want to make it clear that the emission mechanisms of conventional fluorescent OLEDs using TTA and our UC-OLED are different. As we described in Figure 1 and discussed in the introduction, both S1 and T1 are excited after charge recombination in the TTA emitter in conventional fluorescent OLEDs; thus, the theoretical maximum IQE of the device is 62.5% ($=25+75/2$). However, an applied voltage greater than the bandgap energy of the TTA emitter is necessary for excitation; therefore, the maximum power consumption efficiency is 62.5%. In the case of our UC-OLED, the theoretical maximum IQE is 50% ($=100/2$) because only T1 is selectively excited. Simultaneously, the applied voltage is reduced by half because of the TTA process sensitized by the low-energy CT state; therefore, the maximum power consumption efficiency could become 100% if the device could reach the maximum EQE.

3. *For traditional fluorescence device, the luminance lower than 10^4 cd m⁻² is unacceptable. In this work, the luminance of TbPe-doped 1,2-ADN/NDI-HF device is at the scale of 10^3 cd m⁻², which is much lower than the level of TbPe-doped devices. Such as ACS Appl. Mater. Interfaces, 2019, 11, 26. (EQE: 15.3%); J. Mater. Chem. C, 2015, 3, 8834. (EQE: 18.1%).*

Response: First, we want to clarify that the OLED devices in these references use phosphorescent or TADF materials for sensitizing fluorescence from TbPe. These high EQEs cannot be theoretically achieved in OLEDs based on the direct excitation of fluorescent materials. As we discussed in the introduction, phosphorescent and TADF materials have the intrinsic problem of low stability because the high-energy T1 causes their degradation, although they show high EQE. Therefore, conventional fluorescent materials are still used in commercial blue OLEDs. The importance of our findings lies in the reduction of the turn-on voltage using the emitter commonly utilized in commercial OLEDs. A discussion has been added to the abstract, introduction, and conclusion. As discussed on page 10, line 3, we recognize that there is still potential to improve the EQE of our device to close to the theoretical limit. A high luminance of over 10^4 cd/m² would possibly be achieved if the EQE is improved.

4. *There is no sense talking about so-called low turn-on voltage without emphasizing on the efficiency of device. Low turn-on voltage does not represent state-of-the-art device performance. It can also be achieved by external circuit engineering and device structure optimization. Several blue OLEDs exhibit high efficiency along with low turn-on voltage, such as Chem. Eng. J., 2023, 465, 142848. (Von: 2.8 V and EQE: 29.2%); Chem. Eur. J., (Von: 2.9 V and EQE: 20.6%); J. Mater. Chem. C, 2018, 6, 7839. (Von: 2.6 V and EQE: 26.4%). The crucial issue in this work is the unacceptable EQE of devices (< 3.52%), which is not match with the high-quality requirements of Nature Communications. Besides, authors are suggested to revised the title focusing on the scientific issue.*

Response: The turn-on voltage is primarily determined by the energy of the excited state formed after charge recombination. Therefore, there is a limit that can be reached by optimization of the device structure. The references the reviewer raised clearly reflect the fact that the turn-on voltages in these references are all 2.5~2.9 V, and a turn-on voltage of less than 1.5 V for blue emission has never been achieved. We have demonstrated a turn-on voltage of less than 1.5 V for the first time by passing through novel energy transfer steps for blue emission by using the interfacial charge transfer state as an intermediate.

Another point we want to emphasize is that the references the reviewer raised are for blue OLEDs using TADF or phosphorescent materials. As we discussed in the

introduction, these materials have the intrinsic problem of low stability because the high-energy T1 causes their degradation, although they show high EQE. Therefore, conventional fluorescent materials are still used in commercial blue OLEDs. The importance of our findings lies in the reduction of the turn-on voltage using the emitter commonly utilized in commercial OLEDs. A discussion has been added to the abstract, introduction, and conclusion.

5. The authors do not carefully compare analogous literatures with low turn-on voltage. To our knowledge, the turn-on voltages of phosphorescent and TADF devices are less than 3.0 V, respectively. For instance, J. Mater. Chem. C, 2022, 10, 8349. (Von: 2.6 V); Dyes Pigm., 2022, 200, 110041. (Von: 2.4 V); J. Mater. Chem. C, 2021, 9, 10334. (Von: 2.6 V); J. Mater. Chem. C, 2021, 9, 260. (Von: 2.6 V). Moreover, authors should summarize the representative blue electroluminescence devices with turn-on voltages less than 3.0 V and list a table including device structure, turn-on voltages, luminance, efficiency, CIE and citation in Supplementary Information.

Response: We have not included Dyes Pigm., 2022, 200, 110041 because this paper reports a green OLED, but we have added the other three references the reviewer suggested in Figure 4e. Following the reviewer's suggestion, we have added Table S2 summarizing the device parameters for the references used to create Figure 4e. The host/dopant materials in the emission layer (EML), peak wavelength, turn-on voltage, maximum EQE, maximum luminance, and CIE are listed in Table S2. All the references for OLEDs use a device structure with ITO as an anode and bottom emission. The explanation has been added in the caption of Table S2.